# Real Foliage Plants as Visual Stimuli to Improve Concentration and Attention in Elementary Students

**DOI:** 10.3390/ijerph16050796

**Published:** 2019-03-05

**Authors:** Yun-Ah Oh, Seon-Ok Kim, Sin-Ae Park

**Affiliations:** 1Department of Animal and Plant Assisted Therapy, Graduate School of Agriculture and Animal Science, Konkuk University, Seoul 05029, Korea; yunahoh@hanmail.net (Y.-A.O.); kso0804@naver.com (S.-O.K.); 2Department of Environmental Health Science, Sanghuh College of Life Science, Konkuk University, Seoul 05029, Korea

**Keywords:** brain waves, electroencephalography, horticultural therapy, mood state, socio-horticulture

## Abstract

This study was conducted to determine the physiological and psychological benefits of foliage plants as visual stimuli. Twenty-three elementary students (aged 11 to 13 years old) participated in this study. In a crossover design, electroencephalography (EEG) was used to measure and determine the psycho-physiological effects of four different visual stimuli: an actual plant, artificial plant, photograph of a plant, and no plant. Subjective evaluations of emotions were assessed using the profile of mood state and semantic differential methods immediately after exposure to each visual stimulus. A significant decrease in theta waves of the frontal lobe was associated with presentation of the actual plants. This response indicated that the viewing of living plants prompted improvements in the attention and concentration of the elementary students. Furthermore, the presentation of the living plants was associated with more positive mood states, such as feelings of comfort and naturalness. In conclusion, actual plants may improve attention and prompt psychological relaxation in elementary students relative to artificial plants, photographs of plants, or the absence of plants.

## 1. Introduction

Owing to rapid urbanization and industrialization, the number and extent of green spaces have decreased and, as a result, humans have become distanced from Nature. This disconnect can spawn social issues, such as human alienation and emotional deterioration [1]. The savannah theory of Orians and the Wilson’s biophilia hypothesis explain the instinctive preferences of humans for the natural world [2,3]. The desire for closeness to nature has increased, and people have introduced plants into their indoor environments [4,5,6].

Research show that most people seek nature for pleasure, social activities, or for physical exercise [7]. Kaplan and Kaplan suggested that exposure to the natural environment can effectively restore attention [8]. Ulrich found that stress from the external environment can be reduced through exposure to nature [9]; this observation is supported by reports of reductions in cortisol levels and blood pressure and positive mood changes in response to the natural environment [10,11,12,13].

In South Korea, children aged 10–18 years spend an average of 0.62 h outdoors on weekdays, which is very low compared to time spent outdoors by children in Japan, Germany, and the USA [14]. The less children become exposed to the natural environment, the more actively opportunities for children to experience nature should be provided.

Park et al. found that horticultural activities were effective in altering physical, psychological, emotional, behavioral, educational, and cognitive aspects [15]; the effects on emotional intelligence and an eco-friendly attitude were highest in children. In childhood, the body and cognitive functions must develop simultaneously to achieve emotional stabilization [16], and educators have suggested that exposure to nature in childhood is important for emotional development [17,18]. Horticultural activities stimulate a variety of emotions and provide opportunities for communication through group activities, allowing children to express their emotions naturally [19]. Bagot and Corraliza et al. suggested that children feel more positive emotions and have a high resilience to stress in nature-friendly schools [20,21]. Based on the results of various studies demonstrating the benefits of the natural environment on the well-being of humans, interest in the relationship between the natural environment and human health has increased [22].

Previous studies have showed that the viewing of foliage plants can lead to physiological and psychological relaxation by stabilizing the autonomic nervous system and activating alpha brainwaves [23,24,25]. Sympathetic nerve activity and oxyhemoglobin concentration in the left frontal cortex reportedly decreases following the performance of tasks involving foliage plants relative to tasks without plants [26]. Furthermore, transplanting activities using real plants induced better emotional states and lower stress levels than transplanting activities using artificial flowers and computer tasks [27,28]. Moreover, placing foliage plants indoors has been shown to attenuate techno-stress [29]: the psychophysiological consequences of technology-induced alterations of living, such as an increased blood pressure and eye fatigue. Various studies have been conducted on the physiological and psychological changes that occur in adults after visual stimulation with green plants; however, such studies on children are severely limited. This study therefore investigated both the physiological and psychological responses of elementary students after viewing different foliage plants through electroencephalography (EEG) and the evaluation of participants’ emotional responses.

## 2. Materials and Methods

### 2.1. Subjects

The present study recruited 23 elementary school students. Researchers distributed flyers advertising the recruitment on the bulletin boards of community centers near Konkuk University, such as churches and apartment buildings. The inclusion criteria were as follows: right-hand dominance and no pre-existing physical and emotional disorders that could affect the physiological data. The first criterion was informed by reports that brain activation differs between right- and left-handed individuals [30]. Participants received the equivalent of $20 (USD) as an incentive to complete the experiment. This study was approved by the Institutional Review Board of Konkuk University (7001355-2017-10-HR-206).

### 2.2. Experimental Environment

The experiment was conducted in a room at Konkuk University. A desk and chair was placed in the center of a room. The height of the chair was adjusted so that both feet of the subject would reach the ground. To minimize external visual stimulation, white hardboard paper was placed before the desk, and ivory-colored curtains were installed on either side of it (Figure 1a); a 1.8-by-1.6-m space was thus enclosed. A potted plant was placed on the desk at a distance of 0.5 m from the subject (Figure 1b). The average conditions of the experimental space were as follows: temperature, 21.8 °C; humidity, 25.2%; and illumination, 1465.8 lux.

This study employed four visual stimuli: (1) actual plants, (2) artificial plants, (3) photograph of plants, and (4) no plants. As shown in Figure 2, the stimuli were set up as follows: (1) Actual plants: non-patterned *Epipremnum aureum* were planted in a white flower pot (width 55 cm, height 15 cm). (2) Artificial plants: artificial foliage, similar in appearance to *E. aureum*, were arranged in a manner similar to the actual plants and in the same kind of white flower pot. (3) Photograph: researchers printed a life-sized color photograph that was taken of the living plant used as the actual plant stimulus in condition (1). (4) Absence of plants: the same white flower pot used in conditions (2) and (3) was partially filled with plain soil.

### 2.3. Experimental Protocol

Prior to the experiments, the height, weight, and body composition of the participants were measured using an anthropometer (Ok7979; Samhwa, Seoul, Korea) and a body-fat analyzer (ioi 353; Jawon Medical. Gyeongsan, Korea). Demographic information, such as age and sex, was collected through a survey.

A wireless EEG (Quick-20, Cognionics, Inc., San Diego, CA, USA) was attached to the head of the subjects who were then seated in the experimental space for 3 min in order to habituate them to the novel environment. Prior to each trial, the researcher drew lots to determine the order in which to present the four visual stimuli (actual plant, artificial plant, photograph of plant, and no plant). Physiological data were collected for 3 min for each treatment. Before each stimulus was presented, the subjects were asked to look at the white wall in front of them for 1 min to encourage relaxation. After the presentation of each stimulus, the subjects were given a survey regarding their psychological response to the stimulations. The duration of the entire experiment was ~30 min per subject (Figure 3).

### 2.4. Measurement Items

Brain waves from the cerebral cortex are recorded as electrical signals classified as following types based on their frequencies: gamma (30–50 Hz), beta (14–30 Hz), alpha (8–13 Hz), delta (4–8 Hz), and theta (4–8 Hz) [31]. Each type of wave is association with a different situation: gamma, anxiety and excitement; beta, tension; alpha, relaxation; delta, deep sleep; and theta, shallow sleep [32]. In this study, alpha and theta were analyzed to inform assessments of physiological stability and improvement of attention under the four visual conditions.

According to the 10–20 international system of electrode placement, the electrode was attached to the left ear lobe (A1), and the EEG was performed using a total of eight channels: Fp1 (left prefrontal cortex), Fp2 (right prefrontal cortex), F3 (left frontal lobe), F4 (right frontal lobe), P3 (left parietal lobe), P4 (right parietal lobe), O1 (left occipital lobe), and O2 (right occipital lobe) [33]. Only two channels, F3 (left frontal lobe) and F4 (right frontal lobe), were analyzed in this study because the frontal lobe is reportedly associated with rational decision making and judgment (Figure 4).

The profile of mood state (POMS) and semantic differential (SD) methods were used to investigate the psychological reaction of each participant to each stimulus.

The POMS method was developed by McNair et al. as a way of assessing temporary mood or emotional states that vary according to the surrounding conditions of the subjects [34]. The questionnaire used for this method consists of 30 questions that assess tension-anxiety, depression, anger, fatigue, confusion, and vigor [35]; it was translated into Korean by Yeun and Shin-Park. Total mood disorder (TMD) is assessed through the questionnaire. A value corresponds to each response, and lower total scores indicate better emotional states.

The SD method was developed by Osgood as a method of evaluating emotions with adjectives [36]. The questionnaire consists of three categories: comfortable to uncomfortable, natural to artificial, and relaxed to awakening. The participant’s degrees of emotion were scored on a scale of 13 points, and higher scores indicated better emotional states.

### 2.5. Data Analysis

SPSS (Version 22 for Windows; IBM, Armonk, NY, USA) was used to conduct one-way analysis of variance and Kruskal-Wallis tests. A *p*-value of <0.05 indicated statistical significance. Demographic data were analyzed using Microsoft Excel (Office 2016; Microsoft Crop., Redmond, WA, USA) to generate descriptive statistics of the means, standard deviations, and percentages.

## 3. Results

### 3.1. Descriptive Characteristics

The participants (boys: *n* = 9, 39.1%; girls: *n* = 14, 60.9%) in this study were (mean ± SD) 12.2 ± 1.0 years old, 149.6 ± 11.3 cm in height, weighed 44.3 ± 11.8 kg, and their body mass index was 19.4 ± 3.1 kg m^−2^; according to the World Health Organization, these values fall within the normal range for the age-group (Table 1) [37].

### 3.2. Electroencephalography (EEG)

The theta waves, which reportedly evince drowsiness and are used as a measure of low attention or concentration [38], were found to be significantly reduced when participants looked at the actual plants relative to the other stimuli (Table 2). This finding suggests that the improved attention of the participants was attributable to the visual stimulus of the actual plants rather than general attentional stability or arousal. Furthermore, the actual plant did not prompt any changes in alpha waves in the children.

### 3.3. Subjective Mood Assessment

There were no significant differences in the POMS induced by the stimuli; however, a lower TMD was associated with the presentation of the actual plant (Figure 5). This result revealed the potential of actual plants to prompt positive mood states. The SD findings indicated that the subjects felt more comfortable (*p* = 0.049) and natural (*p* = 0.021) when they stared at the living plants (Figure 6).

## 4. Discussion

### 4.1. Electroencephalography (EEG)

While other studies have only investigated the advantages of green plants, the present study compared the effects of green plants with other visual plant-related stimuli whose presentations evoked the way in which they are actually perceived in daily life. As display technologies advance, images are projected with increasingly greater clarity. However, this study showed that only actual plants can significantly affect theta waves. This may indicate that all five senses, and not just sight, contributed to the alteration of neural activity in a way that could not be simulated by the other stimuli [38].

The present study found looking at the actual plants significantly reduced theta waves, suggesting that live plants can improve attention. Park reported that the inhibition of theta waves in 11–15-year-old adolescents contributes heavily to their academic performance [39].

Son reported that the placement of green plants in a room changed the brain waves of the people therein, improved their concentration, and relieved their visual fatigue [40]. Kuo reported that living in a natural environment is closely related to the enhancement of attention [41], and contact with nature has been shown to improve the attention of both adults and children [42,43].

As aforementioned, alpha waves indicate a state of rest or relaxation [32,44]. Prior studies conducted with adult subjects have shown that viewing of green plants increases cerebral activity and has a positive effect on their psycho-physiology [45,46]. Moreover, a different study demonstrated that the latter effect was more potent when viewing actual foliage plants than when viewing images of plants [47]; this finding may be attributable to the activation of parasympathetic nerves and the suppression of the sympathetic nervous system [23,48,49,50]. In addition, when working with plants, activity of parasympathetic nerves increases and the left frontal cortical concentration of oxidized hemoglobin decreases among adults working with plants relative to those that are not [26]. However, in contrast to these findings, the present study found not changes in the alpha waves induced by the actual plants in children.

### 4.2. Subjective Mood Assessment

The SD performed by the present study revealed that the actual plant stimulus induces more positive emotions, such as comfort and feelings of naturalness, than did other stimuli. These effects resemble subjective emotional stabilization, e.g., physiological stability and mood improvement, with which a decrease in right prefrontal cortical oxidative hemoglobin concentration was associated in adults that viewed plants [24]. Park et al. also showed that working with plants increased subjective emotional stability relative to plant-free work [26].

Kaplan and Kaplan suggested that the natural environment is effective in restoring attention in humans [8], and Bringslimark et al. reported that indoor plants are effective in reducing stress and for promoting pain relief [2]. In addition, transplanting activities reportedly reduce more stress and induce more positive mood states than does transferring artificial flowers or performing computer-based tasks [27,28]. The application of horticultural therapy was shown to significantly enhance the positive mood of hospitalized patients [51]. According to Van den Berg et al., negative emotions, such as anger, tension, and depression, induced by a horror movie were decreased by 38% after viewing an image of a nature scene [52].

## 5. Conclusions

This study provided scientific support for the physiological and psychological effects of viewing green foliage in children. Visual stimulation with real plants reduced theta waves, indicative of a lack of concentration. Though the actual plants did not enhance alpha waves, which suggest a relaxed state, the children reported their moods to be more comfortable after viewing the living plants. Thus, the visual stimulation of green foliage plants tended to improve attention and feelings of comfort in elementary school students. This study was subject to two limitations. First, our sample size was small. Future studies should recruit more subjects to analyze more specific physiological mechanisms underlying the visual stimulation of green plants. Moreover, a larger study population would allow for the identification of sex-related differences. Second, we only measured momentary mood changes when the subjects look passively at different plant stimuli.

## Figures and Tables

**Figure 1 ijerph-16-00796-f001:**
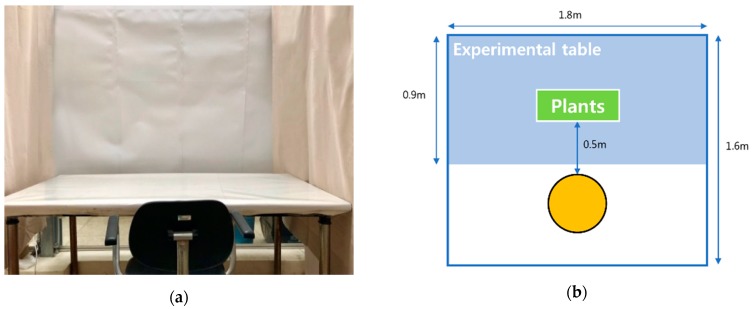
Experimental conditions: (**a**) Photograph of the experimental room; (**b**) Arrangement during the experiment.

**Figure 2 ijerph-16-00796-f002:**
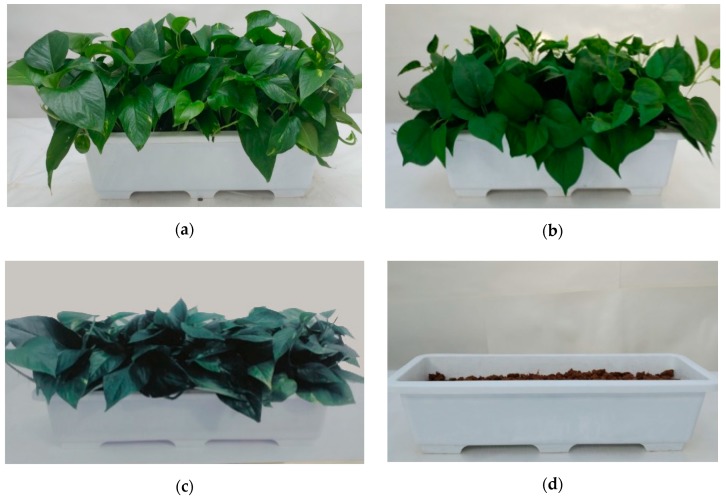
Experimental materials and set-ups used as the visual stimuli: (**a**) Actual plants, (**b**) Artificial plants, (**c**) Photograph of plants, and (**d**) Absence of plants.

**Figure 3 ijerph-16-00796-f003:**
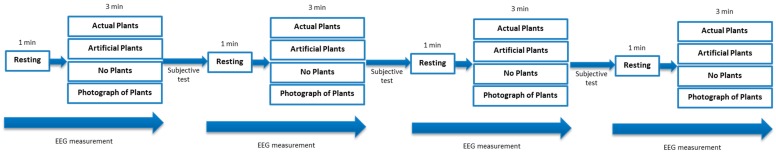
Experimental protocol used to assess the effects of exposure to four stimuli. The responses of the participants were assessed via electroencephalography (EEG) measurements and subjective tests.

**Figure 4 ijerph-16-00796-f004:**
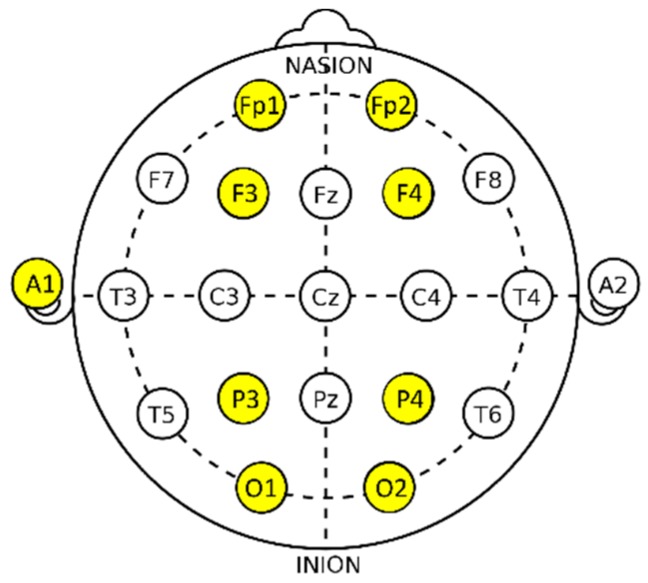
Measuring electrode. Fp: prefrontal cortex; F: frontal lobe; Temporal; C: central; P: parietal; O: occipital; Z: zero, refers to an electrode placed on the midline sagittal plane of the skull; A: the prominent bone process usually found just behind the outer ear.

**Figure 5 ijerph-16-00796-f005:**
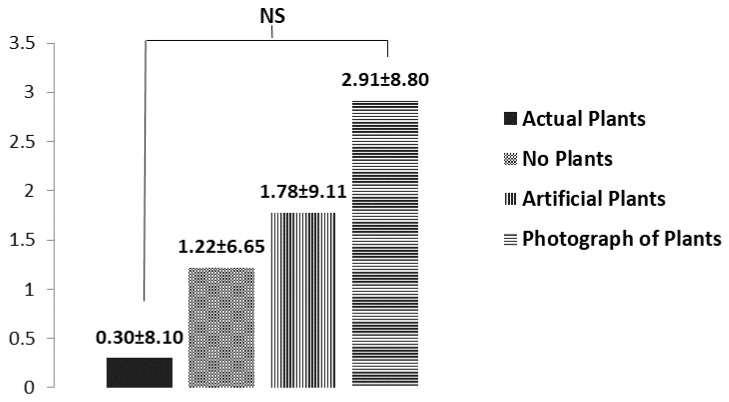
Comparisons of the profile of mood state (POMS) of elementary students in response to four different visual stimuli. Data are presented as means ± standard deviations. NS, non-significant.

**Figure 6 ijerph-16-00796-f006:**
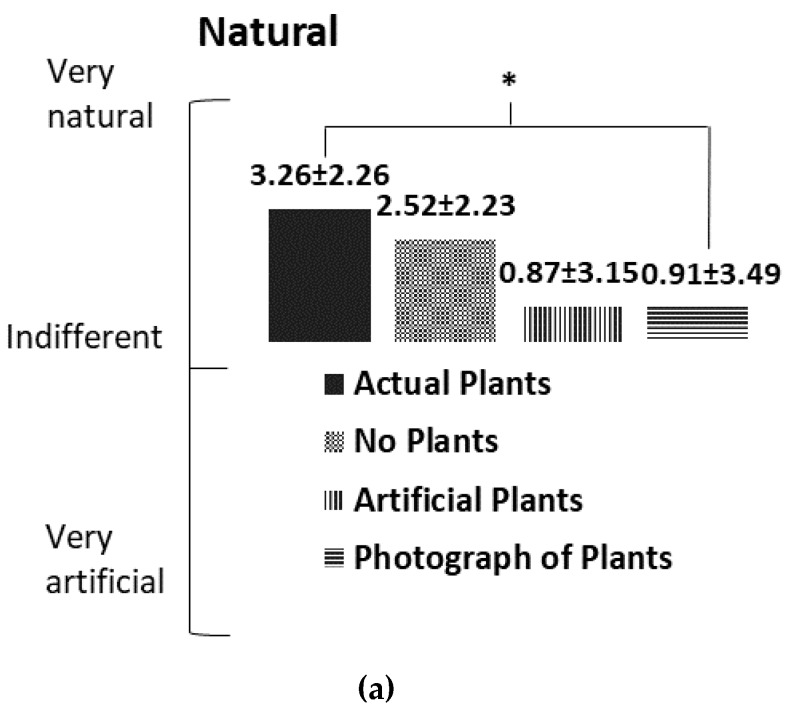
Comparisons of a semantic differential method (SD) of four different visual stimulations. Data are presented as means ± standard deviations. (**a**): natural; (**b**): comfortable; (**c**): relaxation. * *p* < 0.05; NS, non-significant.

**Table 1 ijerph-16-00796-t001:** Descriptive characteristics of the elementary students that participated in the experiment.

Variables	
**Gender**	% (*N*)
**Males**	39.1 (9)
**Females**	60.9 (14)
	Mean (SD)
**Age**	12.2 (1.0)
**Height (cm)**	149.6 (11.3)
**Weight (kg)**	44.3 (11.8)
**Body mass index (kg·m^−2^) ^1^**	19.4 (3.1)

^1^ Body mass index = weight (kg)/height (m^2^).

**Table 2 ijerph-16-00796-t002:** Electroencephalography (EEG) results (alpha and theta waves) following visual stimulation.

EEG	Stimuli	F3 (µV)	F4 (µV)
		Mean ± SD	
Alpha	Actual Plants	0.21 ± 0.79	0.21 ± 0.73
	No Plants	0.22 ± 0.07	0.22 ± 0.07
	Artificial Plants	0.21 ± 0.05	0.20 ± 0.04
	Photograph of Plants	0.20 ± 0.06	0.19 ± 0.06
	Significance	NS	NS
Theta	Actual Plants	0.37 ± 0.94	0.36 ± 0.82
	No Plants	0.38 ± 0.10	0.37 ± 0.11
	Artificial Plants	0.44 ± 0.90	0.44 ± 0.10
	Photograph of Plants	0.43 ± 0.10	0.42 ± 0.11
	Significance	0.028 *	0.048 *

NS: non-significant; * *p* < 0.05.

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
