# Peer review of "Real Foliage Plants as Visual Stimuli to Improve Concentration and Attention in Elementary Students"

_ijerph, 2019, doi:10.3390/ijerph16050796_

Round 1
Reviewer 1 Report
Introduction:
In the introduction you several times make causal relations, this should be re-written, as it is not possible to know if X leads to Y.
Eg. Owing to rapid urbanization and industrialization, the number and extent of green spaces have decreased and, as a result, humans have become distanced from nature. This lack of connect can create social issues, such as human alienation and emotional deterioration [1]. As a result, the desire for closeness to nature has increased which has led to humans introducing plants to their indoor environments
Because the opportunities for children to be exposed to nature are diminishing, the importance of experiencing the natural environment is increasing.
Line 36- 37 You write that: Humans are known to seek green environments for purposes of stress relief, cognitive function improvement, immunity enhancement, and physiological stability
Research show that most people seek nature just for pleasure, social activites or for physical exercise.
Methods:
Line 112-113. Describe the randomization process of the different stimuli.
Figures and tables:
I can not read figure 3, 5 and 6
Table 2. Write the significance found for each of the stimuli, and not just * in the botton line. Further it seems odd to me if you only find significant decrease for viewing plants as 3 of the stimuli have the same decrease of 0.01 point, and viewing plants further has a much higher SD. Can you explain this?
Result and discussion section:
Result and discussion section: These two sections should be changed to two separate sections. And you should discuss your results, also the weaknesses, and not just refer to other research to back them up.
Line 185-187 you make a very strong conclusion based on weak and non significant findings on POMS, this is not justified in the results and should be re-written or erased:
There were no significant differences in the results of the POMS, however, the ‘actual plant’
stimulus led to a lower value of the TMD. This indicated that the viewing of actual plants tended to 187 stimulate positive mood states in the subjects
Conclusion:
The conclusion should be rewritten to reflect the results. Based on the findings the conclusion is too strong. The POMS results were non significant and only 2 of the SD results were significant. Further the significance of the EEG is low.
In the conclusion you write that: This response
indicated that the attention and concentration of the elementary students were improved after viewing living plants.
Attention and concentration are two different cognitive processes, you need to specify them in relation to your findings on EEG.
Author Response
Dear Reviewer,
Thank you very much for your time and efforts. We revised the manuscript by following your comments. We marked as Blue color for the changes in the revised manuscript.
Thank you very much!
1. Introduction: In the introduction you several times make causal relations, this should be re-written, as it is not possible to know if X leads to Y.
Eg. Owing to rapid urbanization and industrialization, the number and extent of green spaces have decreased and, as a result, humans have become distanced from nature. This lack of connect can create social issues, such as human alienation and emotional deterioration [1]. As a result, the desire for closeness to nature has increased which has led to humans introducing plants to their indoor environments
Because the opportunities for children to be exposed to nature are diminishing, the importance of experiencing the natural environment is increasing.
: We changed to
Owing to rapid urbanization and industrialization, the number and extent of green spaces have decreased and, as a result, humans have become distanced from nature. This lack of connect can create social issues, such as human alienation and emotional deterioration[1]. The Savannah theory of Oriansand the Biophilia hypothesis of Wilson explain the instinctive preferences of humans for the natural world[2,3]. The desire for closeness to nature has increased, and people have introduced plants to their indoor environments[4-6].
In South Korea, children aged 10–18 years spend time outdoor for 0.62 h on weekdays, which is very low compared to Japan, Germany, and the USA[14]. As children become less exposed to the natural environment, the opportunities for children to experience nature should be provided.
2. Line 36- 37 You write that: Humans are known to seek green environments for purposes of stress relief, cognitive function improvement, immunity enhancement, and physiological stability
Research show that most people seek nature just for pleasure, social activites or for physical exercise.
: We changed to
Research show that most people seek nature just for pleasure, social activities or for physical exercise [7].
3. Methods: Line 112-113. Describe the randomization process of the different stimuli.
: We changed to
Prior to the experiment, the researcher drew lots and randomly presented the four visual stimuli (actual plants, artificial plants, photograph of plants, or no plants). Physiological data were collected for 3 min for each treatment.
4. Figures and tables: I can not read figure 3, 5 and 6
: We modified to higher resolution figures.
5. Table 2. Write the significance found for each of the stimuli, and not just * in the botton line. Further it seems odd to me if you only find significant decrease for viewing plants as 3 of the stimuli have the same decrease of 0.01 point, and viewing plants further has a much higher SD. Can you explain this?
: We added significance in the table 2.
We think SD can appear high regardless of brainwave reduction because it is a subjective evaluation of each stimulus using self-reporting questionnaires.
6. Result and discussion section:
Result and discussion section: These two sections should be changed to two separate sections. And you should discuss your results, also the weaknesses, and not just refer to other research to back them up.
: We separated Results and Discussion. We also added weaknesses of this study and suggestion of further study in discussion section.
7. Line 185-187 you make a very strong conclusion based on weak and non significant findings on POMS, this is not justified in the results and should be re-written or erased.
There were no significant differences in the results of the POMS, however, the ‘actual plant’
stimulus led to a lower value of the TMD. This indicated that the viewing of actual plants tended to 187 stimulate positive mood states in the subjects
: We changed to
There were no significant differences in the results of the POMS, however, the ‘actual plant’ stimulus led to a lower value of the TMD. This showed the potential to induce positive mood states by looking at actual green plants.
8. Conclusion: The conclusion should be rewritten to reflect the results. Based on the findings the conclusion is too strong. The POMS results were non significant and only 2 of the SD results were significant. Further the significance of the EEG is low.
: We rephrased the conclusion as
This study provided scientific support of the physiological and psychological effects of viewing green foliage in children. The theta waves, which indicate a lack of concentration, were significantly decreased by visual stimulation with real plants. On the other hand, there were no difference in the alpha waves, which are activated in relaxation. Moreover, the subjects showed their mood as comfortable and natural after viewing the living plants. In conclusion, the visual stimulation of green foliage plants showed a tendency to improve attention and led to psychological in elementary school students.
9. In the conclusion you write that: This response
indicated that the attention and concentration of the elementary students were improved after viewing living plants.
Attention and concentration are two different cognitive processes, you need to specify them in relation to your findings on EEG.
: We changed to
In conclusion, the visual stimulation of green foliage plants showed a tendency to improve attention and led to psychological in elementary school students.
Reviewer 2 Report
1) Tecnho-stress (Line 65): This is not a widely recognized term in this context. Recommend rewriting this sentence to clarify/elucidate your meaning.
2) “Right-hand dominant”? (Line 78): Unknown reason for this to be a inclusion criteria. Presumably realted to EEG, but the readers of IJERPH may not know this. Best t ostate explicitly.
3) igures are challenging to read
4) Results and Discussion (Line 148): Recommend separating Results and Discussion sections. Results are objective findings, Discussion is where explanations occur.
5) EEG (Section 3.2) In this section, only theta and alpha waves are discussed. Are other types of waves not measured, or where they measured and found to be not statistically significant, and so not discussed. Either way, the reasoning for either should be explained, either in the results or (probably AND) in Section 2. Possibly in section 1 as part of the intro, for those readers not familiar with EEG.
6) “Indicated” (Line 158) is a stong word. It makes assumptions about the cause of the results. “Suggested” is a more appropriate term.
7) Paragraph in Section 3.2: Recommned moving this to Discussion section
8) Section 3.3: As above, best to separate Results (objective) and Discussion.
9) Conclusions not supported by your data. It is okay that findings were not statsitcally significant. This was a low-powered study. But your statements need to reflect the statistical findings that you have, not the ones you hoped to have.
10) Provide some discussion about the difference between the 3 plant conditions (Actual, Fake, Picture). Why might there be some/no differences between these three exposures for your physiological/psychological metrics?
11) Limitations: Recommend having a limitations section to issues with your study. Both generalizability (which any study has), as well as specific reasons why your findings may have been diffierent than expected. (e.g. low power/small sample, POMS not designed for such frequent repeated use, etc…)
(see attached)

Author Response
Dear Reviewer,
Thank you very much for your time and efforts. We revised the manuscript by following your comments. We marked as Red color for the changes in the revised manuscript.
Thank you very much!
1) Tecnho-stress (Line 65): This is not a widely recognized term in this context. Recommend rewriting this sentence to clarify/elucidate your meaning.
We changed to
: Moreover, techno-stress, which is the negative psychological response due to the used of altered habits of work and home situation such as increasing blood pressure or pulse, and eye fatigue, was reduced by placing the foliage plants indoor.
2) “Right-hand dominant”? (Line 78): Unknown reason for this to be a inclusion criteria. Presumably realted to EEG, but the readers of IJERPH may not know this. Best t ostate explicitly.
We changed to
: Banich et al. that the brain activation was different between right-handed and left-handed. On this basis, the inclusion criteria were as follows: the participants had to be right hand dominant and not have specific physical and emotional disorders that could affect the physiological data.
3) igures are challenging to read: We improved resolution of figures.
4) Results and Discussion (Line 148): Recommend separating Results and Discussion sections. Results are objective findings, Discussion is where explanations occur.
: We separated results section and discussion section.
5) EEG (Section 3.2) In this section, only theta and alpha waves are discussed. Are other types of waves not measured, or where they measured and found to be not statistically significant, and so not discussed. Either way, the reasoning for either should be explained, either in the results or (probably AND) in Section 2. Possibly in section 1 as part of the intro, for those readers not familiar with EEG.
We added
: Brain waves are recording electrical signals from the cerebral cortex, and classify the types into gamma (30-50 Hz), beta (14-30 Hz), alpha (8-13Hz), delta (4-8 Hz), and theta (4-8 Hz) depending on the frequency (Park, 2005). Each waves are activated in different situation-gamma in anxiety and excitement, beta in tension, alpha in relaxation, delta in deep sleep, and theta in shallow sleep (Kim and choi, 2001). In this study, alpha and theta were analyzed to find out about the physiological stability and improvement of the attention by four different visual stimuli of green plants.
6) “Indicated” (Line 158) is a stong word. It makes assumptions about the cause of the results. “Suggested” is a more appropriate term.
We changed to
: This suggested that the visual stimulus of green plants effectively improved the attention of the participants and was a result of looking at the living plants rather than through stability or arousal.
7) Paragraph in Section 3.2: Recommned moving this to Discussion section, 8) Section 3.3: As above, best to separate Results (objective) and Discussion.
: We separated results and discussion section.
9) Conclusions not supported by your data. It is okay that findings were not statsitcally significant. This was a low-powered study. But your statements need to reflect the statistical findings that you have, not the ones you hoped to have.
We rewrote the conclusions as
: This study provided scientific support of the physiological and psychological effects of viewing green foliage in children. The theta waves, which indicate a lack of concentration, were significantly decreased by visual stimulation with real plants. On the other hand, there were no difference in the alpha waves, which are activated in relaxation. Moreover, the subjects showed their mood as comfortable and natural after viewing the living plants. In conclusion, the visual stimulation of green foliage plants showed a tendency to improve attention and led to psychological in elementary school students.
10) Provide some discussion about the difference between the 3 plant conditions (Actual, Fake, Picture). Why might there be some/no differences between these three exposures for your physiological/psychological metrics?
We added
: This might be because the activation of the brain could be different depends on the five senses. The development of projection technology enables visualize images with greater clarity, but it couldn’t stimulate other four senses except visual sense.
11) Limitations: Recommend having a limitations section to issues with your study. Both generalizability (which any study has), as well as specific reasons why your findings may have been diffierent than expected. (e.g. low power/small sample, POMS not designed for such frequent repeated use, etc…)
We added limitations at the last paragraph of discussion (4.2., 4.3.).
This study has limitation of the small number of samples. Further studies require more subjects to analyze more specific physiological mechanisms of the visual stimulation of green plants through the comparison between males and females.
In this study, we only measured momentary mood changes when the subjects passively look at different plant stimuli. Also, there were no significantly difference in POMS. This may be because POMS questionnaire is not sensitive enough to be used four times within 30 minutes.
Round 2
Reviewer 1 Report
1. The conclusion in the abstract has not been revised. This needs to be revised to reflect the findings and conclusion in the paper.
Furthermore, the results showed that actual plants led to positive mood
23 states in the participants. In conclusion, actual plants were more effective as visual stimuli in
24 improving concentration and causing psychological relaxation in elementary students, when
25 compared with artificial plants, photographs of plants, or no plants.
2. I would still want to see the significance levels for the change in all conditions in table 3, and not only for the actual plants.
3. I still can not read figure 3, 5 and 6, which makes it difficult to comment on. However, it seems to me that it would make more sense if figure 5 show both the pre-and post measure and significance levels, as you refer to the change in TMD in the text as a positive result. This also accounts for figure 6.
There were no significant differences in the results of the POMS, however, the ‘actual plant’
185 stimulus led to a lower value of the TMD.
4. the text you have added thoughout the paper, seems stangely added and does not relate to the following text.
Eg.: In this study, the Ɵ waves were significantly reduced when looking at the actual living plants
195 and it indicated the improvement of the attention by viewing the actual green. This might be
196 because the brain waves could be different depends on the five senses[39]. The development of
197 projection technology enables visualize images with greater clarity, but it couldn’t stimulate other
198 four senses except visual sense. Park reported that the inhibition of theta waves in 11–15-year-old
199 adolescents is very important for their academic achievements [38].
200 Son reported that when green plants were placed in a room, the
5. The following sentence needs to be discussed and nuanced, as your results only show that they are more attentive when looking at the actual green plant, but not if they will then afterwards concentrate better in their academic work.
In summary, viewing of green plants can
206 improve a child's ability to concentrate.
6. a word is missing in the conclusion, further I still find it a too strong conclusion on psychological effect based on only two significant measures, when the POMS showed no significance.
7. a separate section on limitation of the study is recommended.
8. the discussion section, does not entail discussion, simply a presentation of results found by others and your results. You need to actually critically and nuanced discuss yours and others results.
9.the English language in your revisions need extensive editing.
Author Response
Dear Reviewer,
Thank you very much for your time and efforts. We revised the manuscript by following your comments. We marked as Green color for the changes in the revised manuscript.
Thank you very much!
1. The conclusion in the abstract has not been revised. This needs to be revised to reflect the findings and conclusion in the paper.
Furthermore, the results showed that actual plants led to positive mood23 states in the participants. In conclusion, actual plants were more effective as visual stimuli in
24 improving concentration and causing psychological relaxation in elementary students, when
25 compared with artificial plants, photographs of plants, or no plants
: We changed to
Furthermore, the presentation of the living plants was associated with more positive mood states, such as feelings of comfort and naturalness. In conclusion, actual plants may improve attention and prompt psychological relaxation in elementary students relative to artificial plants, photographs of plants, or the absence of plants.
2. I would still want to see the significance levels for the change in all conditions in table 3, and not only for the actual plants.
: There are no table 3 in this paper, but only table 2. Because this statistical method used a nonparametric on-way distribution analysis method, the level of method is not obtained.
3. I still can not read figure 3, 5 and 6, which makes it difficult to comment on. However, it seems to me that it would make more sense if figure 5 show both the pre-and post measure and significance levels, as you refer to the change in TMD in the text as a positive result. This also accounts for figure 6.
: We modified to higher resolution figures. Also, figure 5 and 6 showed a comparison between post-processing evaluations, not pre and post stimulation comparison.
4. the text you have added thoughout the paper, seems stangely added and does not relate to the following text.
Eg.: In this study, the Ɵ waves were significantly reduced when looking at the actual living plants
195 and it indicated the improvement of the attention by viewing the actual green. This might be
196 because the brain waves could be different depends on the five senses[39]. The development of
197 projection technology enables visualize images with greater clarity, but it couldn’t stimulate other
198 four senses except visual sense. Park reported that the inhibition of theta waves in 11–15-year-old
199 adolescents is very important for their academic achievements [38].
200 Son reported that when green plants were placed in a room, the
: We changed to
While other studies have only investigated the advantages of green plants, the present study compared the effects of green plants with other visual plant-related stimuli whose presentations evoked the way in which they are actually perceived in daily life. As display technologies advance, images are projected with increasingly greater clarity. However, this study showed that only actual plants can significantly affect theta waves. This may indicate that all five senses, and not just sight, contributed to the alteration of neural activity in a way that could not be simulated by the other stimuli [38].
5. The following sentence needs to be discussed and nuanced, as your results only show that they are more attentive when looking at the actual green plant, but not if they will then afterwards concentrate better in their academic work.
In summary, viewing of green plants can
206 improve a child's ability to concentrate.
: We changed to
Similarly, this study also showed a significant decrease in the theta waves in response to viewing actual green plants. In summary, viewing of green plants can improve a child's ability to attention.
6. a word is missing in the conclusion, further I still find it a too strong conclusion on psychological effect based on only two significant measures, when the POMS showed no significance.
: We changed to
Thus, the visual stimulation of green foliage plants tended to improve attention and feelings of comfort in elementary school students.
7. a separate section on limitation of the study is recommended.
: We added the limitation section and wrote as
This study was subject to two limitations. First, our sample size was small. Future studies should recruit more subjects to analyze more specific physiological mechanisms underlying the visual stimulation of green plants. Moreover, a larger study population would allow for the identification of sex-related differences. Second, we only measured momentary mood changes when the subjects look passively at different plant stimuli.
8. the discussion section, does not entail discussion, simply a presentation of results found by others and your results. You need to actually critically and nuanced discuss yours and others results.
: We added
While other studies have only investigated the advantages of green plants, the present study compared the effects of green plants with other visual plant-related stimuli whose presentations evoked the way in which they are actually perceived in daily life. As display technologies advance, images are projected with increasingly greater clarity. However, this study showed that only actual plants can significantly affect theta waves. This may indicate that all five senses, and not just sight, contributed to the alteration of neural activity in a way that could not be simulated by the other stimuli [38].
9.the English language in your revisions need extensive editing.
: We improved the English style and checked grammatical errors. We did a proofreading of English to native experts.
Reviewer 2 Report
Recommend having a native English speaker edit, there are multiple grammatical errors in the updated version
Author Response
Dear Reviewer,
Thank you very much for your time and efforts.
1) Recommend having a native English speaker edit, there are multiple grammatical errors in the updated version
: We improved the English style and checked grammatical errors. We did a proofreading of English to native experts.